# GRAPH BARLOW TWINS: A SELF-SUPERVISED REPRESENTATION LEARNING FRAMEWORK FOR GRAPHS

## ABSTRACT

The self-supervised learning (SSL) paradigm is an essential exploration area, which tries to eliminate the need for expensive data labeling. Despite the great success of SSL methods in computer vision and natural language processing, most of them employ contrastive learning objectives that require negative samples, which are hard to define. This becomes even more challenging in the case of graphs and is a bottleneck for achieving robust representations. To overcome such limitations, we propose a framework for self-supervised graph representation learning – *Graph Barlow Twins*, which utilizes a cross-correlation-based loss function instead of negative samples. Moreover, it does not rely on non-symmetric neural network architectures – in contrast to state-of-the-art self-supervised graph representation learning method *BGRL*. We show that our method achieves as competitive results as the best self-supervised methods and fully supervised ones while requiring fewer hyperparameters and substantially shorter computation time (ca. 30 times faster than BGRL).

## 1 INTRODUCTION

Graph representation learning has been intensively studied for the last few years, having proposed various architectures and layers, like GCN (Kipf & Welling, 2017), GAT (Veličković et al., 2018), GraphSAGE (Hamilton et al., 2017) etc. A substantial part of these methods was introduced in the semi-supervised learning paradigm, which requires the existence of expensive labeled data (e.g. node labels or whole graph labels). To overcome this, the research community has been exploring unsupervised learning methods for graphs. This resulted in a variety of different approaches including: shallow ones (DeepWalk (Perozzi et al., 2014), Node2vec (Grover & Leskovec, 2016), LINE (Tang et al., 2015)) that ignore the feature attribute richness, focusing only on the structural graph information; and graph neural network methods (DGI (Veličković et al., 2019), GAE, VGAE (Kipf & Welling, 2016)) that build representations upon node or graph features, achieving state-of-the-art performance in those days.

Recently self-supervised paradigm is the most emerging branch of unsupervised graph representation learning and gathers current interest and strenuous research effort towards better results. The most prominent methods were developed around the contrastive learning approach, such as GCA (Zhu et al., 2020b), GraphCL (You et al., 2020), GRACE (Zhu et al., 2020a) or DGI (Veličković et al., 2019). Although contrastive methods are popular in many machine learning areas, including computer vision and natural language processing, their fundamental limitation is the need for negative samples. Consequently, the sampling procedure for negative examples highly affects the overall quality of the embeddings. In terms of images or texts, the definition of negative samples might seem not that problematic, but in the case of graphs there is no clear intuition. For instance, what is the negative counterpart for a particular node in the graph, should it be a node that is not a direct neighbor, or a node that is in a different graph component? There are multiple options available, but the right choice strictly dependent on the downstream task.

Researchers have already tackled the problem of building so-called *negative-sample-free* methods. In research being conducted in computer vision they obtained successful results with methods such as BYOL (Grill et al., 2020), SimSiam (Chen & He, 2020) or Barlow Twins (Zbontar et al., 2021). These models utilize siamese network architectures with various techniques, like gradient stopping, asymmetry or batch and layer normalizations, to prevent collapsing to trivial solutions. Based on

BYOL, Thakoor et al. (2021) proposed the Bootstrapped Representation Learning on Graphs (BGRL) framework. It utilizes two graph encoders: an online and a target one. The former one passes the embedding vectors to a predictor network, which tries to predict the embeddings from the target encoder. The loss is measured as the cosine similarity and the gradient is backpropagated only through the online network (gradient stopping mechanism). The target encoder is updated using an exponential moving average of the online encoder weights. Such setup has been shown to produce graph representation vectors that achieve state-of-the-art performance in node classification using various benchmark datasets. Notwithstanding, assuming asymmetry between the network twins (such as the predictor network, gradient stopping, and a moving average on the weight updates) the method is conceptually complex.

Employing a symmetric network architecture would seem more intuitive and reasonable, hence in this paper, we propose **Graph Barlow Twins (G-BT)**, a self-supervised graph representation learning framework, which computes the embeddings cross-correlation matrix of two distorted views of a single graph. The approach was firstly introduced in image representation learning as the Barlow Twins model (Zbontar et al., 2021) but was not able to handle graphs. The utilized network architecture is fully symmetric and does not need any special techniques to build non trivial embedding vectors. The distorted graph views are passed through the same encoder which is trained using the backpropagated gradients (in a symmetrical manner).

Our main contributions can be summarized as follows: (I) We propose a self-supervised graph representation learning framework Graph Barlow Twins. It is built upon the recently proposed Barlow Twins loss, which utilizes the embedding cross-correlation matrix of two distorted views of a graph to optimize the representation vectors. Our framework neither requires using negative samples (opposed to most other self-supervised approaches) nor it introduces any kind of asymmetry in the network architecture (like state-of-the-art BGRL). Moreover, our architecture is converges substantially faster than all other state-of-the-art methods. (II) We evaluate our framework in node classification tasks: (1) for 5 smaller benchmark datasets in a transductive setting, (2) using the ogb-arxiv dataset from the Open Graph Benchmark (also in the transductive setting), (3) for multiple graphs in the inductive setting using the PPI (Protein-Protein Interaction) dataset, and finally (4) for the large-scale graph dataset ogb-products in the inductive setting. We use both GCN-based encoders as well as a GAT-based one. We observe that our method achieves analogous results compared to state-of-the-art methods. (III) We ensure reproducibility by making the code of both our models as well as experimental pipeline available (currently attached in the supplementary materials).

## 2 RELATED WORKS

**Self-supervised learning** The idea of self-supervised learning (SSL) has a long history. Introduced in the early work of Schmidhuber (Schmidhuber, 1990) has more than 30 years of exploration and research now. Recently self-supervised learning was again rediscovered and found a broad interest, especially in computer vision and natural language processing. One of the most prominent SSL methods for image representation learning, Bootstrap Your Own Latent, BYOL (Grill et al., 2020), performs on par or better than the current state of the art on both transfer and semi-supervised benchmarks. It relies on two neural networks that interact and learn from each other. From an augmented view of an image, it trains the first one to predict the target network representation of the same image under a different view. At the same time, it updates the second network with a slow-moving average of the first network. Another approach to image representation SSL implements simple siamese networks, namely SimSiam (Chen & He, 2020). It achieves comparative results while not demanding negative samples, large batches, nor momentum encoders. Authors emphasize collapsing solutions for the loss and structure but show how a stop-gradient operation plays an essential role in preventing it. Recent method, Barlow Twins (Zbontar et al., 2021), advances the SSL field with a new objective function that naturally avoids collapses by measuring the cross-correlation matrix between the outputs of two twin, identical networks fed with distorted versions of a sample, and makes it as close to the identity matrix as possible. Representations of distorted versions of samples are then expected to be similar, reducing the redundancy between them. What differentiates the method is that it does not require large batches or asymmetry between the network twins. It outperforms previous methods on ImageNet for semi-supervised classification.

**Graph representation learning** Learning the representation also spreads to other domains. The graph embedding problem has also attracted much attention from the research community worldwide in recent years. Plenty of methods have been developed, each focused on a different aspect of network embeddings, such as proximity, structure, attributes, learning paradigm or scalability. There exist plenty of shallow methods, among others DeepWalk (Perozzi et al., 2014), Node2vec (Grover & Leskovec, 2016) or LINE (Tang et al., 2015), that use a simple notion of graph coding through random walks or on encoder-decoder objectives that optimize first and second-order node similarity. More complex graph neural networks, such as GCN(Kipf & Welling, 2017) or GraphSAGE (Hamilton et al., 2017) implements the basic encoder algorithm with various neighborhood aggregation. Following the extension, graph attention network GAT (Veličković et al., 2018) leverages masked self-attentional layers to address the shortcomings of graph convolutions and their troublesome approximations.

**Self-supervised graph representation learning** Inspired by the success of contrastive methods in vision and NLP, the procedures were also adapted to graphs. Early DGI (Veličković et al., 2019) employs GNN to learn node embeddings and obtains the graph embedding via a readout function and maximizes the mutual information between node embeddings and the graph embedding by discriminating nodes in the original graph from nodes in a corrupted graph. GCA (Zhu et al., 2020b) studied various augmentation procedures. GRACE (Zhu et al., 2020a) creates two augmented versions of a graph, pulls together the representation of the same node in both graphs, and pushes apart representations of every other node. Recent GraphCL (You et al., 2020) method is another example of representative approach using contrastive learning. All the previous methods use negative sampling approaches for the embedding optimization, yet such setting has a high complexity. To overcome this, BGRL (Thakoor et al., 2021) proposed to use an approach that does not rely on negative samples. It uses two kinds of encoder networks (online and target), introducing a non-intuitive asymmetric pipeline architecture, but provides state-of-the-art SSL results. Moreover, it relies on several techniques to prevent trivial solutions (gradient stopping, momentum encoder). A concurrent approach to BGRL is DGB (Che et al., 2020).

## 3 PROPOSED FRAMEWORK

Motivated by the emerging self-supervised learning paradigm and its recent applications in graph representation learning (BGRL (Thakoor et al., 2021)), we propose **Graph Barlow Twins** – a framework that builds node embeddings using a symmetric network architecture and an empirical cross-correlation based loss function. The overall pipeline of our framework is shown in Figure 1. The consecutive processing steps can be described as follows:

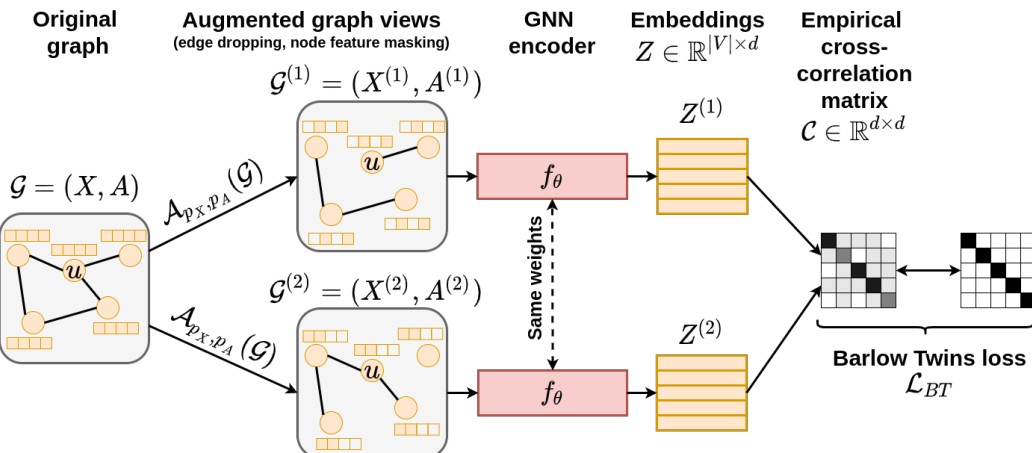

Figure 1: Overview of our proposed Graph Barlow Twins framework. We transform an input graph $\mathcal{G}$ using an augmentation function and obtain two views: $\mathcal{G}^{(1)}$ and $\mathcal{G}^{(2)}$. We pass both of them through the same GNN encoder $f_\theta$ to compute two embedding matrices $Z^{(1)}$, $Z^{(2)}$. We build a loss function such that the embeddings' empirical cross-correlation matrix $\mathcal{C}$ is optimized into the identity matrix.

**Graph data** We represent a graph $\mathcal{G}$ with nodes $\mathcal{V}$ and edges $\mathcal{E}$ as the tuple: $(X, A)$, where $X \in \mathbb{R}^{|\mathcal{V}| \times k}$ is the node feature matrix and $k$ is the feature dimensionality; $A \in \{0,1\}^{|\mathcal{V}| \times |\mathcal{V}|}$ is the adjacency matrix, such that $A_{i,j} = 1$ iff $(i, j) \in \mathcal{E}$. In the general case, a graph could also have associated edge features or graph level features, but for simplicity we omit those here. Nevertheless, these could also be used in our framework, as long as the encoder can make use of such features.

**Generating graph views via augmentation** Following other works (Thakoor et al., 2021; Zhu et al., 2020b; You et al., 2020; Zhu et al., 2020a), we select two kinds of augmentations – *edge dropping* and *node feature masking* – and generate two views of the input graph $\mathcal{G}^{(1)}$ and $\mathcal{G}^{(2)}$. In the edge dropping case, we remove edges according to a generated mask of size $|\mathcal{E}|$ (number of edges in the graph) with elements sampled from the Bernoulli distribution $\mathcal{B}(1 - p_A)$. When it comes to masking node features, we employ a similar scheme and generate a mask of size $k$ also sampled from the Bernoulli distribution $\mathcal{B}(1 - p_X)$. Note that we mask node features at the scale of the whole graph, i.e. the same features are masked for each node. Other works apply different augmentation parameters $p_X, p_A$ for each generated view, but as our framework is fully symmetrical, we postulate that it is enough to use the same parameters to generate both augmentations (see Section 4.4).

**Encoder network** The main component of the proposed framework is the encoder network $f_\theta : \mathcal{G} \to \mathbb{R}^{|\mathcal{V}| \times d}$. It takes an augmented graph as the input and computes (in our case) a $d$-dimensional representation vector for each node in the graph. Note that we do not specify any particular encoder network and one may use even encoders that construct embeddings for edges or whole graphs. In our experiments, we will show the application of GCN (Kipf & Welling, 2017) and GAT (Veličković et al., 2018) based encoder networks. Both augmented graph views $\mathcal{G}^{(1)}, \mathcal{G}^{(2)}$ are passed through the same encoder, resulting in two embedding matrices $Z^{(1)}$ and $Z^{(2)}$, respectively. The original Barlow Twins method specified also a projector network (implemented as an MLP) to reduce high embedding dimensionality (of the ResNet encoder). Our approach eliminates that step as it uses GNNs with low dimensional embeddings.

**Loss function** In our work, we propose to use a *negative-sample-free* loss function to train the encoder network. We first normalize the embedding matrices $Z^{(1)}$ and $Z^{(2)}$ along the batch dimension (a mean of zero and a standard deviation equal to one), and then we compute the empirical cross-correlation matrix $\mathcal{C} \in \mathbb{R}^{d \times d}$:

$$\mathcal{C}_{ij} = \frac{\sum_b Z_{b,i}^{(1)} Z_{b,j}^{(2)}}{\sqrt{\sum_b (Z_{b,i}^{(1)})^2} \sqrt{\sum_b (Z_{b,j}^{(2)})^2}}, \tag{1}$$

where $b$ are the batch indexes and $i, j$ are the indexes of embeddings. Such setting was originally proposed under the name **Barlow Twins**. Neuroscientist H. Barlow's *redundancy-reduction principle* has motivated many methods both in supervised and unsupervised learning (Deco & Parra, 1997; Schmidhuber et al., 1996; Ballé et al., 2017). Recently, Zbontar et al. (2021) has employed this principle to build a self-supervised image representation learning algorithm (we bring this idea to the domain of graph-structured data).

The cross-correlation matrix $\mathcal{C}$ is optimized by the Barlow Twins loss function $\mathcal{L}_{\mathrm{BT}}$ (see Equation 2) to be equal to the identity matrix. The loss is composed of two parts: (I) the invariance term and (II) the redundancy reduction term. The first one forces the on diagonal elements $\mathcal{C}_{ii}$ to be equal to one, hence making the embeddings invariant to the applied augmentations. The second term optimizes the off-diagonal elements $\mathcal{C}_{ij}$ to be equal to zero – this results in decorrelated components of the embedding vectors.

$$\mathcal{L}_{\mathrm{BT}} = \sum_i (1 - \mathcal{C}_{ii})^2 + \lambda \sum_i \sum_{j \neq i} \mathcal{C}_{ij}^2 \tag{2}$$

The $\lambda > 0$ parameter defines the trade-off between the invariance and redundancy reduction terms when optimizing the overall loss function. In Tsai et al. (2021), the authors proposed to use $\lambda = \frac{1}{d}$, which we employ in our experimental setting. Otherwise, one can perform a simple grid search to find the best $\lambda$ value in a particular experiment.

Please note, that in such setting the gradient is symmetrically backpropagated through the encoder network. We do not rely on any special techniques, like momentum encoders, gradient stopping, or predictor networks. In preliminary experiments, we also investigated the Hilbert-Schmidt Independence Criterion (due to its relation to the Barlow Twins objective (Tsai et al., 2021)), but we did not observe any performance gain.

# 4 EXPERIMENTS

We evaluate the performance of our model using a variety of popular benchmark datasets, including smaller ones, such as WikiCS, Amazon-Photo or Coauthor-CS, as well as larger ones, such as ogb-arxiv, ogb-products, provided by the Open Graph Benchmark (Hu et al., 2020). We describe all datasets along with their statistics in Appendix A. In this section, we will provide an overview experimental scenario details, and the discussion of the results. Overall, we use a similar experimental setup, as the state-of-the-art self-supervised graph representation learning method BGRL (Thakoor et al., 2021), so we can perform a fair comparison to this method. To track our experiments and provide a simple way for reproduction, we employ the Data Version Control tool (DVC; Kuprieiev et al. (2021)). We perform all experiments on a TITAN RTX GPU with 24GB RAM.

## 4.1 EVALUATION PROTOCOL

**Self-supervised framework training** We start the evaluation procedure by training the encoder networks using our proposed **Graph Barlow Twins** framework. In all scenarios, we use the AdamW optimizer (Gugger & Howard, 2018) with weight decay equal to $10^{-5}$. The learning rate is updated using a cosine annealing strategy with a linear warmup period. Our framework uses a single set of augmentation parameters for both graph views. Therefore we do not use reported values of these parameters from other publications that use two different sets. Instead we perform a grid search over the range: $p_A, p_X : \{0, 0.1, \ldots, 0.5\}$ for 500 epochs with a warmup time of 50 epochs. We implement our experiment using the PyTorch Geometric (Fey & Lenssen, 2019) library. All datasets are available in this library as well. The details about the used augmentation hyperparameters, node embedding dimensions and the encoder architecture are given in Appendix B.

**Node embedding evaluation** We follow the linear evaluation protocol proposed by Veličković et al. (2019). We use the trained encoder network, freeze the weights and extract the node embeddings for the original graph data without any augmentations. Next, we train a $L_2$-regularized logistic regression classifier from the Scikit learn (Pedregosa et al., 2011) library. We also perform a grid search over the regularization strength using following values: $\{2^{-10}, 2^{-9}, \ldots, 2^9, 2^{10}\}$. In the case of the larger ogb-arxiv, ogb-products and the PPI datasets, the Scikit implementation takes too long to converge. Hence, we implement the logistic regression classifier in PyTorch and optimize it for 1000 steps using the AdamW optimizer. We check various weight decay values using a smaller grid search: $\{2^{-10}, 2^{-8}, \ldots, 2^8, 2^{10}\}$. We use these classifiers to compute the classification accuracy and report mean and standard deviations over 20 model initializations and splits, except for the ogb-arxiv, ogb-products and PPI datasets, where we there is only one data split provided – we only re-initialize the model weights 20 times (5 times for ogb-products due to long training time).

## 4.2 TRANSDUCTIVE EXPERIMENTS

We evaluate and compare our framework to other graph representation learning approaches on 6 real-world datasets using the transductive setting. The whole graph including all the node features is observed during the encoder training. The node labels (classes) are hidden at that moment (unsupervised learning). Next, we use the frozen embeddings and labels of training nodes to train the logistic regression classifier.

### 4.2.1 SMALL AND MEDIUM SIZES BENCHMARK DATASETS

Our first experiment uses 5 small and medium sized popular benchmark datasets, namely: WikiCS, Amazon Computers, Amazon Photos, Coauthor CS and Coauthor Physics.

**Encoder model** Similarly to (Thakoor et al., 2021), we build our encoder $f_\theta$ as a 2-layer GCN Kipf & Welling (2017) network. After the first GCN layer we apply a batch normalization layer (with momentum equal to 0.01) and the PReLU activation function. Accordingly to the original Barlow Twins method, we do not apply any normalization or activation to the final layer. A graph convolution layer (GCN) uses the diagonal degree matrix $\mathbf{D}$ to apply a symmetrical normalization to the adjacency matrix with added self-loops $\hat{\mathbf{A}} = \mathbf{A} + \mathbf{I}$. Hence the propagation rule of such layer is defined as follows:

$$\text{GCN}(X, A) = \hat{\mathbf{D}}^{-\frac{1}{2}} \hat{\mathbf{A}} \hat{\mathbf{D}}^{-\frac{1}{2}} X \mathbf{W} \tag{3}$$

Note that we do not include the activation $\sigma(\cdot)$ in this definition, as we first apply the batch normalization and then the activation function.

**Results and discussion** We train our framework for a total of 1000 epochs, but we observe that our model converges earlier at about 500-900 epochs (depending on the dataset; see Appendix B). This is significantly faster than the state-of-the-art method BGRL, which converges and reports results for 10 000 epochs. Additionally, we reproduce the results of BGRL and provide values for BGRL at 1000 epochs. In Table 1 we report the mean node classification accuracy along with the standard deviations. As our experimental scenario was aligned with BGRL one, we re-use their reported scores and compare them to our results. We observe that our proposed method outperforms other baselines and achieves comparable results to state-of-the-art methods. Moreover, our G-BT model outperforms BGRL at 1000 epochs.

Table 1: Mean and std accuracy of transductive node classification over 20 data splits and initializations obtained in BGRL paper (Thakoor et al., 2021) and our experiment ($*$) within the same experimental setup. *OOM* denotes running out of memory on a 16GB V100 GPU.

| | **WikiCS** | **Am-CS** | **Am-Photo** | **Co-CS** | **Co-Physics** |
|---|---|---|---|---|---|
| Raw features | $71.98 \pm 0.00$ | $73.81 \pm 0.00$ | $78.53 \pm 0.00$ | $90.37 \pm 0.00$ | $93.58 \pm 0.00$ |
| DeepWalk | $74.35 \pm 0.06$ | $85.68 \pm 0.06$ | $89.44 \pm 0.11$ | $84.61 \pm 0.22$ | $91.77 \pm 0.15$ |
| DeepWalk + fts | $77.21 \pm 0.03$ | $86.28 \pm 0.07$ | $90.05 \pm 0.08$ | $87.70 \pm 0.04$ | $94.90 \pm 0.09$ |
| DGI | $75.35 \pm 0.14$ | $83.95 \pm 0.47$ | $91.61 \pm 0.22$ | $92.15 \pm 0.63$ | $94.51 \pm 0.52$ |
| MVGRL | $77.52 \pm 0.08$ | $87.52 \pm 0.11$ | $91.74 \pm 0.07$ | $92.11 \pm 0.12$ | $95.33 \pm 0.03$ |
| GRACE (10k epochs) | $80.14 \pm 0.48$ | $89.53 \pm 0.35$ | $92.78 \pm 0.45$ | $91.12 \pm 0.20$ | OOM |
| BGRL (10k epochs) | $79.36 \pm 0.53$ | $89.68 \pm 0.31$ | $92.87 \pm 0.27$ | $93.21 \pm 0.18$ | $95.56 \pm 0.12$ |
| **G-BT**$*$ ($\leq$1k epochs) | $76.65 \pm 0.62$ | $88.14 \pm 0.33$ | $92.63 \pm 0.44$ | $92.95 \pm 0.17$ | $95.07 \pm 0.17$ |
| BGRL$*$ (1k epochs) | $73.24 \pm 0.62$ | $87.37 \pm 0.40$ | $91.77 \pm 0.57$ | $92.07 \pm 0.06$ | OOM$*$ |
| GCA | $78.35 \pm 0.05$ | $88.94 \pm 0.15$ | $92.53 \pm 0.16$ | $93.10 \pm 0.01$ | $95.73 \pm 0.03$ |
| Supervised GCN | $77.19 \pm 0.12$ | $86.51 \pm 0.54$ | $92.42 \pm 0.22$ | $93.03 \pm 0.31$ | $95.65 \pm 0.16$ |

### 4.2.2 OGB-ARXIV DATASET

In the next experiment, we use ogb-arxiv – a larger graph from the Open Graph Benchmark (Hu et al., 2020) with about 170 thousand nodes and about 1.1 million edges.

**Encoder model** Due to the larger size of the graph, we extend the encoder $f_\theta$ to a 3-layer GCN model. We employ batch normalization and PReLU activations after the first and second layer, leaving the final layer as is (i.e. without any activation of normalization). In the BGRL paper, the authors suggested to use layer normalization together with weight standardization (Qiao et al., 2019), yet we did not observe any performance gain, but more importantly the training procedure was unstable, with many peaks in the loss function.

**Results and discussion** In Table 2 we report the mean classification accuracy along with the standard deviations. Note that we provide values for both validation and test splits, as the provided data splits are build according to chronological order. Hence, any model will be more affected with the out-of-distribution error on further (in time) away data samples. We evaluate our model for 500 epochs but it converges as fast as about 300-400 epochs (further training did not give any

improvements). The model achieves results which are only 1.5 pp off to the state-of-the-art method BGRL, which in turn takes 10 000 epochs to converge to such solution.

## 4.3 INDUCTIVE EXPERIMENTS

We evaluate our proposed **G-BT** framework in inductive tasks over a single and multiple graphs.

### 4.3.1 PPI

For the inductive learning case with multiple graphs, we employ the **Protein-Protein Interaction (PPI)** dataset (Zitnik & Leskovec, 2017). Aligned with other methods, we provide results for multilabel node classification in terms of the Micro-F1 score.

**Encoder model** We employ a Graph Attention (GAT) Veličković et al. (2018) based encoder model, as previous works have shown better results of such network compared to standard GCN layers on PPI. Specifically, we build our encoder $f_\theta$ as a 3-layer GAT network with skip connections. The first and second layer uses 4 attention heads of size 256 which are concatenated, and the final layer uses 6 attention heads

Table 2: Mean and std accuracy of transductive node classification the **ogb-arxiv** dataset over 20 data splits and initializations obtained in BGRL paper (Thakoor et al., 2021) and our experiment (∗) within the same experimental setup.

|  | **Validation** | **Test** |
|---|---|---|
| MLP | $57.65 \pm 0.12$ | $55.50 \pm 0.23$ |
| node2vec | $71.29 \pm 0.13$ | $70.07 \pm 0.13$ |
| DGI | $71.26 \pm 0.11$ | $70.34 \pm 0.16$ |
| GRACE (10k epochs) | $72.61 \pm 0.15$ | $71.51 \pm 0.11$ |
| BGRL (10k epochs) | $72.53 \pm 0.09$ | $71.64 \pm 0.12$ |
| **G-BT**∗ (300 epochs) | $71.16 \pm 0.14$ | $70.12 \pm 0.18$ |
| Supervised GCN | $73.00 \pm 0.17$ | $71.74 \pm 0.29$ |

of size 512, whose outputs are averaged instead of applying concatenation. In the GAT model, an attention mechanism learns the weights that are used to aggregate information from neighboring nodes. The attention weights $\alpha_{ij}$ are computed according to the following equation:

$$\alpha_{ij} = \frac{\exp\left(\text{LeakyReLU}\left(\mathbf{a}^T\left[\mathbf{W}h_i||\mathbf{W}h_j\right]\right)\right)}{\sum_{k\in\mathcal{N}_i}\exp\left(\text{LeakyReLU}\left(\mathbf{a}^T\left[\mathbf{W}h_i||\mathbf{W}h_k\right]\right)\right)} \tag{4}$$

where $\mathcal{N}_i$ are the neighbors of node $i$, $\mathbf{W}$ is a trainable matrix to transform node attributes, $\mathbf{a}$ is the trainable attention matrix, and $||$ denotes the concatenation operation.

We use the ELU activation for the first and second layer, leaving the last layer without any activation function. We do not apply any normalization techniques in the model as preliminary experiments showed no performance improvement.

**Results and discussion** We train our framework using a batch size of 1 graph for a total of 500 epochs, which turned out to be enough for the model to converge (we conducted some preliminary experiments). In Table 3, we report the mean Micro-F1 score along with the standard deviations over 20 model initialization, as this dataset provided only one data split. Training for only 500 epochs provided results on par with SOTA method – BGRL – our model achieves $70.49$ using a GAT encoder.

### 4.3.2 OGB-PRODUCTS

We study the applicability of our proposed model in the case of large-scale graphs. We select the ogb-products dataset, which has about 2.5 million nodes and 61 million edges.

**Encoder model and setup** We utilize the same GAT-based encoder as for PPI. Due to the size of this dataset and the resulting training time, we decide to perform inductive node classification, i.e., during training we use only the nodes from the training set and edges among them. Moreover, as this graph does not fit into GPU memory, we selected a batched setting with neighbor sampling (as proposed in Hamilton et al. (2017)) instead of the full-batch scenario. We train our model with a batch size of 512 for 100 epochs.

**Results and discussion** BGRL does not report results for this dataset, so we modify the implementation of the BGRL method to accept batches instead of whole graphs and evaluate it on this dataset. We also include results from the OGB leaderboard, but note that virtually all methods reported there are trained in a semi-supervised setting, contrary to our approach in the self-supervised setting. Therefore, we may expect worse results. We summarize the mean and std node classification accuracy values in Table 4. We observe that G-BT highly outperforms BGRL on both validation and test sets.

Table 3: Mean and std Micro-F1 of multilabel node classification the **PPI dataset** over 20 model initializations obtained in BGRL paper (Thakoor et al., 2021) and our experiment (∗) within the same experimental setup.

|  | PPI (test set) |
|---|---|
| Raw features | 42.20 |
| DGI | $63.80 \pm 0.20$ |
| GMI | $65.00 \pm 0.02$ |
| GRACE | $66.20 \pm 0.10$ |
| GRACE GAT encoder (1k epochs) | $69.71 \pm 0.17$ |
| BGRL GAT encoder (1k epochs) | $70.49 \pm 0.05$ |
| **G-BT**∗ (500 epochs) | $70.49 \pm 0.19$ |
| Supervised GAT | $97.30 \pm 0.20$ |

Table 4: Mean and std accuracy of inductive node classification on the **ogb-products** dataset over 5 model initializations obtained in the OGB leaderboard and our experiment (∗) within the same experimental setup.

|  | Validation | Test |
|---|---|---|
| Features∗ | $63.18 \pm 0.01$ | $50.93 \pm 0.01$ |
| DeepWalk∗ | $87.42 \pm 0.09$ | $73.11 \pm 0.44$ |
| DeepWalk + fts∗ | $87.84 \pm 0.09$ | $73.38 \pm 0.11$ |
| BGRL∗ (100 epochs) | $78.06 \pm 2.12$ | $63.97 \pm 1.62$ |
| **G-BT**∗ (100 epochs) | $85.04 \pm 0.23$ | $70.46 \pm 0.38$ |
| Supervised GCN | $92.00 \pm 0.03$ | $75.64 \pm 0.21$ |

### 4.4 Augmentation hyperparameter sets

In our model, we postulate to use the same augmentation function hyperparameters to generate both graph views. This is motivated by the symmetrical architecture of our model, and hyperparameter search complexity. Performing a simple grid search over the value space yields in our case a total number of $6^2 = 36$ evaluated combinations (values: $\{0, 0.1, \ldots, 0.5\}$). In contrary, usage of different parameter sets for both graph views, would generate $(6^2)^2 = 1296$ combinations, which can be further reduced by exploiting the symmetrical architecture, yielding a final value of $630 = \binom{36}{2}$ combinations to evaluate. To demonstrate the impact of using both the same and different augmentation hyperparameter value sets we provide the results in Figure 2. There is no substantial difference in terms of test accuracy.

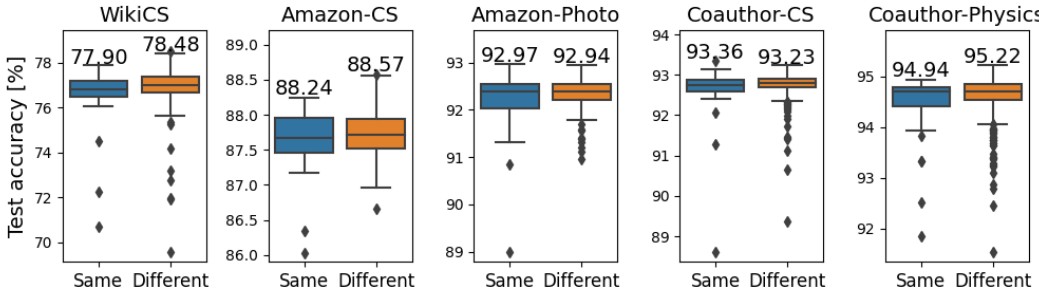

Figure 2: Comparison of using the "Same" and "Different" augmentation hyperparameter sets.

### 4.5 Training time comparison

We compare the training time of all considered models by the duration of single epoch (the evaluation phase is the same in all models). We run each model for 10 training epochs and report the mean and standard deviation of the time measurements (Table 5). In virtually all cases our model takes the least

time for a single training iteration due to the simple symmetrical architecture. Compared to BGRL our method speeds up computations about 17-42 times.

Table 5: Single epoch running time (in seconds) averaged over 10 training epochs.

|  | WikiCS | Am-CS | Am-Photo | Co-CS | Co-Phy |
|---|---|---|---|---|---|
| DeepWalk | $0.83 \pm 0.02$ | $0.96 \pm 0.02$ | $0.62 \pm 0.02$ | $1.22 \pm 0.03$ | $2.25 \pm 0.03$ |
| DGI | $0.06 \pm 0.03$ | $0.09 \pm 0.00$ | $0.05 \pm 0.00$ | $0.19 \pm 0.00$ | $0.50 \pm 0.21$ |
| MVGRL | OOM | OOM | OOM | OOM | OOM |
| GRACE | $0.33 \pm 0.10$ | $0.43 \pm 0.03$ | $0.15 \pm 0.01$ | OOM | OOM |
| BGRL | $0.12 \pm 0.01$ | $0.18 \pm 0.01$ | $0.08 \pm 0.00$ | $0.35 \pm 0.01$ | OOM |
| Epochs \| training time [s]: | 10 000 \| 1 200 | 10 000 \| 1 800 | 10 000 \| 800 | 10 000 \| 3 500 | 10 000 \| - |
| G-BT | $0.05 \pm 0.00$ | $0.07 \pm 0.00$ | $0.04 \pm 0.00$ | $0.22 \pm 0.05$ | $0.44 \pm 0.01$ |
| Epochs \| training time [s]: | 900 \| 45 | 600 \| 42 | 500 \| 20 | 900 \| 198 | 900 \| 396 |
| Speedup (vs BGRL): | **26x** | **42x** | **40x** | **17x** | - |

## 4.6 BATCHED PROCESSING

Our proposed method allows working in both full-batch and mini-batch settings. For most considered datasets, splitting them up into batches is not required as these fit completely into the GPU's memory. Nevertheless, we run additional experiments where we train our G-BT model on these datasets in a batched manner. Batches are created using neighbor sampling, i.e. for a $k$-layer encoder model, we sample the $k$-hop neighborhood of a node. More specifically, we first subsample the direct neighbors, then we sample neighbors of those, etc (as proposed in Hamilton et al. (2017)). We re-use the augmentation hyperparamter values and number of epochs found in the full-batch case and retrain the G-BT model for each batch size 5 times (Table 6). We observe an expected decrease in performance when using the batched scenario (subject to further finetuning).

Table 6: Evaluation of G-BT model in batched setting.

| Batch size | WikiCS | Am-CS | Am-Photo | Co-CS | Co-Phy |
|---|---|---|---|---|---|
| Full-batch | $76.65 \pm 0.62$ | $88.14 \pm 0.33$ | $92.63 \pm 0.44$ | $92.95 \pm 0.17$ | $95.07 \pm 0.17$ |
| 256 | $75.69 \pm 1.02$ | $87.93 \pm 0.39$ | $91.24 \pm 0.46$ | $91.82 \pm 0.22$ | $94.98 \pm 0.14$ |
| 512 | $75.83 \pm 0.64$ | $88.21 \pm 0.44$ | $91.21 \pm 0.44$ | $91.62 \pm 0.22$ | $94.95 \pm 0.12$ |
| 1024 | $75.79 \pm 0.77$ | $87.94 \pm 0.50$ | $91.24 \pm 0.47$ | $91.54 \pm 0.31$ | $94.91 \pm 0.12$ |
| 2048 | $75.58 \pm 0.52$ | $87.92 \pm 0.29$ | $91.21 \pm 0.40$ | $91.43 \pm 0.28$ | $94.84 \pm 0.11$ |

## 5 CONCLUSIONS

In this work we presented Graph Barlow Twins, a self-supervised graph representation learning framework, which utilizes the embeddings' cross-correlation matrix computed from two distorted views of a particular graph. The framework is fully symmetric and does not need any special techniques to build non trivial embedding vectors. It builds representations that are invariant to the applied augmentations and reduces the redundancy in the representation vectors by enforcing the cross-correlation matrix to be equal to the identity matrix (Barlow Twins loss). Using 8 real-world datasets we evaluate our model in node classification tasks, both transductive and inductive, and achieve results that are on par or better than SOTA methods in SSL graph representation learning. We also show that our model converges an order of magnitude faster than other approaches.

Our method allows to reduce the computation cost (faster convergence) keeping a decent performance in downstream tasks. Consequently, it can be used to process larger graph datasets and efficiently perform tasks such as node classification, link prediction or graph classification. These tasks have crucial impact on various machine learning areas where graph structured data is used, e.g. detection of bots or hate speech in social networks, or building graph based recommendation engines.

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

## A  DATASET DESCRIPTIONS

We provide brief descriptions for each dataset, including the basic statistics (see Table 7) and the employed dataset split type for the node classification downstream task:

- **WikiCS** (Mernyei & Cangea, 2020) is a network of Computer Science related Wikipedia articles with edges denoting references between those articles. Each article belongs to one of 10 subfields (classes) and has features computed as averaged GloVe embeddings of the article content. We use the provided 20 train/val/test data splits without any modifications.

- **Amazon Computers, Amazon Photos** (McAuley et al., 2015) are two networks extracted from Amazon's co-purchase data. Nodes are products and edges denote that these products were often bought together. Each product is described using a Bag-of-Words representation (node features) based on the reviews. There are 10 and 8 product categories (node classes), respectively. For these datasets there are no data splits available, so similar to BGRL, we generate 20 random train/val/test splits (10%/10%/80%).

- **Coauthor CS, Coauthor Physics** are two networks extracted from the Microsoft Academic Graph (Sinha et al., 2015). Node are authors and edges denote a collaboration of two authors. Each author is described by the keywords used in their articles (Bag-of-Words representation; node features). There are 15 and 5 author research fields (node classes), respectively. Similarly to the Amazon datasets there are no data splits provided, so we generate 20 random train/val/test splits (10%/10%/80%).

- **ogb-arxiv** is a larger graph from the Open Graph Benchmark (Hu et al., 2020) with about 170 thousand nodes and about 1.1 million edges. The graph was extracted from the Microsoft Academic Graph (Sinha et al., 2015), where nodes represents a Computer Science article on the arXiv platform and edges denote citations across papers. The node features are build as word2vec embeddings of the whole article content. There are 40 subject areas a node can be classified into (node label/class). The ogb-arxiv dataset provides a single train/val/test split, so we use it without any modifications, but we retrain the whole framework 20 times.

- **Protein-Protein Interaction (PPI)** (Zitnik & Leskovec, 2017) consists of 24 separate graphs. Each node in a single graph represents a protein, described by 50 biological features, and edges denote interactions among those proteins. There are 121 node labels, but note that contrary to other cases, PPI uses multilabel classification, i.e. a single protein can be assigned with multiple labels. Aligned with other methods, we provide results in terms of the Micro-F1 score. For PPI, there exists a predefined data split, where 20 graphs are used for training, 2 graphs for validation and 2 graphs for testing. Note that the validation and test graphs are completely unobserved during training time, hence the model is more challenged during inference time.

- **ogn-products** is a large-scale graph from the Open Graph Benchmark (Hu et al., 2020) with about 2.4 million nodes and 62 million edges. The graph was extracted from the Amazon product co-purchasing network. Nodes represent products from the Amazon store and edges denote whether two products were bought together. There are 100 node features, which are obtained from bag-of-words products descriptions reduced using PCA. Each product (node) can be classified into one of 47 categories (node labels). This dataset comes with a predefined data split, so we use as is.

## B  EXPERIMENTAL DETAILS

**Augmentation hyperparameters**  Our proposed framework uses a single pair of augmentation hyperparameters $p_A \in \mathbb{R}, p_X \in \mathbb{R}$ compared to other methods that use different values to generate both graph views. We show that a single set is enough to achieve a decent performance in a symmetrical network architecture like ours. Therefore, we cannot use the reported values of other

Table 7: Dataset statistics. We use small to medium sized standard datasets together with the larger ogb-arxiv dataset in the transductive setting. We also evaluate the inductive setting using the ogb-products and PPI (multiple graphs) dataset.

| Name | Nodes | Edges | Features | Classes |
|------|-------|-------|----------|---------|
| **WikiCS** | 11,701 | 216,123 | 300 | 10 |
| **Amazon Computers** | 13,752 | 245,861 | 767 | 10 |
| **Amazon Photos** | 7,650 | 119,081 | 745 | 8 |
| **Coauthor CS** | 18,333 | 81,894 | 6,805 | 15 |
| **Coauthor Physics** | 34,493 | 247,962 | 8,415 | 5 |
| **ogb-arxiv** | 169,343 | 1,166,243 | 128 | 40 |
| **PPI (24 graphs)** | 56,944 | 818,716 | 50 | 121 (multilabel) |
| **ogb-products** | 2,449,029 | 61,859,140 | 100 | 47 |

works. We instead perform a grid search over these hyperparameters and use those where the model performs the best (in terms of classification accuracy or Micro-F1 score, for PPI). We do not evaluate the model during training and just use the final version after training. We use the following setting:

- the framework is trained to **500 epochs**,

- we set the learning rate warmup time to **50 epochs**,

- for both hyperparameters $p_A$ and $p_X$ we check following values: $\{0, 0.1, \ldots, 0.5\}$.

For values greater than $0.5$ the augmentation removes too much information from the graph. In the case of the ogb-products dataset, due to its large size, we trained our model only for 10 epochs with a warmup period of 2 epochs, but we evaluated the same augmentation hyperparameter values. We summarize the augmentation hyperparameters of the best performing models in Table 8.

Table 8: Augmentation hyperparameters. Ogb-products was trained in the batched setting with a batch size of 512.

| | **G-BT** | |
|------|------|------|
| | $p_A$ | $p_X$ |
| **WikiCS** | 0.2 | 0.1 |
| **Amazon-CS** | 0.4 | 0.1 |
| **Amazon-Photo** | 0.0 | 0.5 |
| **Coauthor-CS** | 0.5 | 0.1 |
| **Coauthor-Physics** | 0.1 | 0.4 |
| **ogb-arxiv** | 0.2 | 0.0 |
| **PPI** | 0.1 | 0.1 |
| **ogb-products**∗ | 0.2 | 0.1 |

**Training setup**  For all datasets, we train our framework using the AdamW (Gugger & Howard, 2018) optimizer with a weight decay of $10^{-5}$. The learning rate is adjusted using a cosine annealing strategy with a linear warmup period up to the base learning rate. During training we set a total number of epochs and an evaluation interval, after which the frozen embeddings are evaluated in downstream tasks (using either the $l_2$ regularized logistic regression from Scikit learn (Pedregosa et al., 2011) with liblinear solver, or the custom PyTorch version with AdamW for ogb-arxiv and PPI). For instance, if we set the total number of epochs to 1000 and the evaluation interval to 500, the model will be evaluated at epochs: 0, 500 and 1000 (three times in total). We report the values for the best performing model found during those evaluations. We summarize these training statistics in Table 9.

Table 9: Training hyperparameters.

| | total epochs | warmup | evaluation interval | base learning rate | best model found at |
|---|---|---|---|---|---|
| | | | **G-BT** | | |
| **WikiCS** | 1 000 | 100 | 100 | $5 * 10^{-4}$ | 900 |
| **Amazon-CS** | 1 000 | 100 | 100 | $5 * 10^{-4}$ | 600 |
| **Amazon-Photo** | 1 000 | 100 | 100 | $1 * 10^{-4}$ | 500 |
| **Coauthor-CS** | 1 000 | 100 | 100 | $1 * 10^{-5}$ | 900 |
| **Coauthor-Physics** | 1 000 | 100 | 100 | $1 * 10^{-5}$ | 900 |
| **ogb-arxiv** | 500 | 100 | 100 | $1 * 10^{-3}$ | 300 |
| **PPI** | 500 | 50 | 100 | $5 * 10^{-3}$ | 500 |
| **ogb-products** | 100 | 10 | 10 | $1 * 10^{-3}$ | 100 |

**Encoder architecture**  We compare our framework against the state-of-the-art self-supervised graph representation learning method BGRL (Thakoor et al., 2021). To provide a fair comparison, we use similar encoder architectures to the ones presented in their paper. We do not use any predictor networks in our framework, so we need to slightly modify the encoders to be better suited for the loss function (as given in the Barlow Twins paper (Zbontar et al., 2021)), i.e. we do not apply any normalization (like batch or layer normalization) or activation function in the final layers of the encoder. Note that the lack of predictor network and batch normalization in the final layer, reduces the overall number of trainable network parameters. In all cases, we use a batch normalization with the momentum equal to $0.01$ (as in BGRL (Thakoor et al., 2021), where they use the equivalent weight decay equal to $0.99$).

For the small up to medium sized datasets, i.e. WikiCS, Amazon-CS, Amazon-Photo, Coauthor-CS, Coauthor-Physics, we use a 2-layer GCN (Kipf & Welling, 2017) based encoder with the following architecture:

- GCN($k, 2d$),
- BatchNorm($2d$),
- PReLU(),
- GCN($2d, d$),

where $k$ is the number of node features and $d$ is the embedding vector size.

For the ogb-arxiv dataset, we use a slightly larger model – a 3-layer GCN (Kipf & Welling, 2017) based encoder. We tried to utilize weight standarization (Qiao et al., 2019) and layer normalization, but our model did not benefit from those techniques (as it helped in BGRL (Thakoor et al., 2021)). The training procedure under this setting was unstable with various fluctuations and peaking of the loss function. The final architecture is summarized as follows:

- GCN($k, d$),
- BatchNorm($d$),
- PReLU(),
- GCN($d, d$),
- BatchNorm($d$),
- PReLU(),
- GCN($d, d$).

In the inductive experiment with the PPI dataset, we use a 3-layer GAT (Veličković et al., 2018) based encoder. Graph Attention network are known to perform better on this dataset compared to GCNs. This was also showed in BGRL (Thakoor et al., 2021), where their approach with GAT layers provided state-of-the-art performance in self-supervised graph representation learning for PPI. Our architecture can be summarized as follows:

- GAT($k$, 256, heads=4) + Linear($k$, $4 * 256$)
- ELU(),
- GAT($4 * 256$, 256, heads=4) + Linear($4 * 256$, $4 * 256$)
- ELU(),
- GAT($4 * 256$, $d$, heads=6) + Linear($4 * 256$, $d$)

The outputs of the attention heads in the first and second layer are concatenated and for the last GAT layer, the attention heads outputs are averaged. In every layer, we utilize skip connections using linear layers to project the outputs of the previous layer (features in the case of the first layer) to the desired dimensionality.

The exact values for the input feature dimension $k$ and the embedding dimension $d$ are given in Table 10.

Table 10: Encoder layer size parameters.

| | **G-BT** | |
| | number of node features $k$ | embedding dimensionality $d$ |
| --- | --- | --- |
| **WikiCS** | 300 | 256 |
| **Amazon-CS** | 767 | 128 |
| **Amazon-Photo** | 745 | 256 |
| **Coauthor-CS** | 6 805 | 256 |
| **Coauthor-Physics** | 8 415 | 128 |
| **ogb-arxiv** | 128 | 256 |
| **PPI** | 50 | 512 |
| **ogb-products** | 100 | 128 |

**Code and reproducibility**    We implement all our models using the PyTorch-Geometric library (Fey & Lenssen, 2019). The experimental pipeline is built using the Data Version Control tool (DVC (Kuprieiev et al., 2021)). It enables to run all experiments in a single command and ensure better reproducibility. We attach the code in the supplementary material. To reproduce the whole pipeline run: `dvc repro` and to execute a single stage use: `dvc repro <stage name>`. There are following stages:

- `preprocess_dataset@<dataset_name>` – downloads the `<dataset_name>` dataset; if applicable, generates the node splits for train/val/test,
- `full_batch_hps@<dataset_name>` – runs the augmentation hyperparameter search for a given dataset in the full-batch case,
- `full_batch_train@<dataset_name>`, `batched_train@<dataset_name>` – trains and evaluates the G-BT model for a given dataset in the full-batch case and the batched scenario, respectively,
- `batched_hps_ogbn_products` – runs the augmentation hyperparameter search for the ogb-products dataset in the batched scenario,
- `batched_train_ogbn_products` – trains and evaluated the G-BT model for the ogb-products dataset in the batched scenario,
- `compare_augmentation_hyperparameter_sets` – loads all full-batch augmentation hyperparameter results, compares the case when using the same or different sets of hyperparameters to generate both graph views (outputs Figure 2),
- `compare_running_times` – computes the average running time of a training epoch for the following methods: DeepWalk, DGI, MVGRL, GRACE, BGRL and G-BT,
- `train_bgrl_full_batch@<dataset_name>` – trains and evaluates the BGRL model in the full-batch case for WikiCS, Amazon-CS, Amazon-Photo, and Coauthor-CS,

- `bgrl_hps_batched@ogbn-products` – runs the augmentation hyperparameter search for BGRL using the ogb-products dataset,

- `bgrl_batched_train@ogbn-products` – trains and evaluates the BGRL model for the ogb-products dataset,

- `evaluate_features_products` – evaluates the performance of ogb-products' raw node features,

- `evaluate_deepwalk_products` – evaluates the performance of DeepWalk on the ogb-products dataset; additionally the case of DeepWalk features concatenated with raw node features is also evaluated.

All hyperparameters described in this Appendix are stored in configuration files in the `experiments/configs/` directory, whereas the experimental Python scripts are placed in the `experiments/scripts/` directory.

