# OpenReview forum: "Graph Barlow Twins: A self-supervised representation learning framework for graphs"
_ICLR.cc/2022/Conference — ICLR 2022 Submitted_

### Official Review · Reviewer_bHed · 2021-10-30

**Correctness:** 4
**Technical Novelty And Significance:** 2
**Empirical Novelty And Significance:** 1
**Recommendation:** 5
**Confidence:** 5

**Main Review:**

This paper is easy to follow, well-written and organized. This paper is a heuristic attempt to apply Barlow Twins in graph domain. Both transductive and inductive experiments are done to evaluate the performance. The proposed method achieved results on par with the SOTA methods while the time and space complexity are lower.

I have some minor concerns below for the authors to address.

1. The novelty of this paper is limited. The challenges to be tackle of the application about Barlow Twins in graph domain is unclear to me.

2. Data augmentation in vision tasks comes from strong human prior, e.g., random resize, cropping and horizontal flipping would not change the semantic of an image. While the graph data augmentation methods used in this paper is borrowed from previous literatures, it makes no sense to me. For example, applying edge dropping to a protein would obviously lead to different bio-molecules.

3. The experimental results are on par with baseline methods on the most tasks. Considering that the low time and space complexity is coming from previous literature, i.e., Barlow Twins, the experimental contribution is limited.

4. In terms of the encoder network and augmentation hyperparameter design, the paper did not provide comprehensive analysis or ablation studies.

5. The authors carefully describe the downstream datasets. Maybe I miss it, but I don't find the pre-trained dataset used in the experiment.

**Summary Of The Paper:**

This paper proposed a self-supervised learning framework for graph representation learning based on a cross-correlation-based loss function. In the proposed framework, two views of the input graph obtained by augmentation methods are passed through the same encoder to compute two embedding matrices, then Barlow Twins loss is used to compute the loss according to the embedding matrices.
The main contribution of this paper lies in that it adapted Barlow Twins from vision to graph representation learning field and evaluated the performance of this self-supervised framework in multiple node classification tasks. The proposed method achieved analogous results compared to SOTA methods with lower time and space complexity.


**Summary Of The Review:**

This paper adapted the recent Barlow Twins to self-supervised graph representation learning and provided some informative empirical experiment results. With such interesting trials, the reviewer expected to see the concerns are well addressed.

---

> ### Author Response · Authors · 2021-11-12
> **Addressing your concerns (1/2)**
>
> Dear Reviewer,
>
> Thank you for your review. We would like to address your concerns, as stated below:
>
> **1. Challenges of Barlow Twins in graph domain**
>
> Let us clarify the challenges of the application of Barlow Twins in the graph domain.
>
> The only way to realistically access the paper’s novelty, in this case, is to compare it with the original Barlow Twins paper. The comparison should consider the research question, methodology, and results of both articles.  The research questions are different – we wanted to answer whether one could utilize a symmetric negative-sample free loss function to obtain expressive embeddings at a fraction of the training time of state-of-the-art graph SSL methods. The methodology seems similar, as it appears in all graph SSL papers nowadays. The obtained results are aligned to the nature of different problems that are solved.
>
> Due to our best knowledge, our paper is a first attempt to utilize a symmetric cross-correlation-based cost function (currently known under the name “Barlow Twins” loss) in the graph representation learning domain.
>
> Let us discuss in detail all new significant proposals that differentiate it from the previous work:
> “Barlow Twins: Self-Supervised Learning via Redundancy Reduction.” In general, the similarity to that work is limited to the utilized cost function. All other computational steps in our proposal are different due to another class of objects we aim at obtaining representation. We introduced a new appropriate encoder, data augmentation functions, simplified part of neural network structure that is not necessary to apply in graph case, and provided experimental results for the batched scenario.
>
> **I. Network encoder structure and its relevancy to the graph data**
>
> The original paper was dealing with vision data. In this domain, any proposed learning model should be invariant with respect to the underlying translational symmetry group (e.g., in image classification, the class should not depend on the object location on that image). In the case of the original Barlow Twins method, this issue was solved by employing the ResNet architecture (in fact, using convolutional layers with pooling operations). In the case of graphs, we care about invariance to an entirely different symmetry group, namely the permutation group (of the node ordering). We implement it using the Graph Convolution and Graph Attention encoder models. Yet it is not automatically known whether such encoder swap would work and provide meaningful representations in the first place. Moreover, in terms of images, we generally assume that each data point (image) is independent (known as the IID property). In the case of graphs, nodes (which we learn representations for), are dependent on other nodes (via edges). Using the batched setting from the original Barlow Twins method might be straightforwardly applicable in our case, as we are passing the whole graph (or a subsampled batch). Nevertheless, our experiments provided more insight to that matter and turned out that it is possible.
>
> **II. Augmentation functions**
>
> The original Barlow Twins paper proposed using a set of well-known augmentation functions for the image domain, i.e. “random cropping, resizing to 224 × 224, horizontal flipping, color jittering, converting to grayscale, Gaussian blurring, and solarization”. None of these is applicable to the graph domain. We had to find suitable augmentation functions and evaluate them. The space of such functions is vast, but as stated in our paper, we wanted to compare our proposed method to BGRL. Hence we used the same augmentation functions as in their paper, namely, edge dropping and node feature masking.
>
> Another modification in our paper is related to the probabilities of the augmentation functions. Both the original Barlow Twins and BGRL papers propose to utilize different probabilities to generate both augmented data views. We proposed to simplify the hyperparameter space and verified experimentally that it is enough to use the same probabilities to generate both views. Such a setting significantly reduces the hyperparameter space search and provides the same results.
>
> **III. Simplification of the model structure**
>
> Original Barlow Twins method for images required a projector network to deal with the dimensionality of the ResNet encoder output (2048). Surprisingly, the method kept improving with an increase of projector network dimensionality. In our proposed architecture, we proposed to completely omit the projector network and work directly with the embedding vectors computed by the encoders. In preliminary experiments, we verified that the use of a projector network does not improve the downstream task results. Moreover, removing the projector network noticeably reduces the number of parameters and learning time.

---

> ### Author Response · Authors · 2021-11-12
> **Addressing your concerns (2/2)**
>
> **IV. Full-batch vs Batched processing**
>
> A popular strategy in graph representation learning is processing the whole graph at once (full-batch). This limits the applications of GNNs to small and medium-sized graphs. Nevertheless, our main competitor BGRL utilized only this scenario in their experiments. In our case, we wanted to show the expressiveness of our proposed model for large-scale graphs (ogb-products dataset). We proposed to adopt a graph batching strategy to our model (through neighbor sampling strategy). Moreover, we reproduced BGRL for the batched scenario.
>
> **V. Novelty in experimental results**
>
> Our experiments provide results for the first graph-related application of a model with a symmetric cross-correlation-based cost function. Moreover, it empirically confirms that results for node classification tasks with λ=1/d perform well (according to Tsai et al. 2021).
>
> We strongly believe that all the provided arguments will convince the reviewer to change the reception of the paper and will consider the change of the paper’s assessment. We stand for the fact that it is hardly possible to obtain better tradeoff results (time-accuracy), as in our paper, without introducing any novelty to the ones previously published.
>
> **Summary**
>
> In summary, our contribution compared to Barlow Twins original paper includes: a new appropriate encoder, data augmentation functions, and a simplified part of neural network structure that is not necessary in the graph case and provide experimental results for the batched scenario.  All of the mentioned differences will be highlighted in the revised version of the manuscript.
>
>
> **2. Data augmentation in graphs**
>
> Indeed, we agree that the employed augmentation functions may not be the best choice for certain types of graphs. Especially in the case of the Protein-Protein-Interaction dataset (PPI), as you correctly noticed, edge dropping may remove edges that are important from the perspective of the protein classification task. Nevertheless, we want to underline that our paper is not focused on surveying different augmentation functions in self-supervised graph representation learning (for more details we would like to direct to “An Empirical Study of Graph Contrastive Learning” by Zhu et al.). One of our subgoals was to compare against the currently best performing graph SSL method - BGRL. To provide a fair comparison, we employed the same augmentation functions as in their paper. Additionally, although the application of node feature masking and edge dropping might not be fully supportive in all possible cases, it still provides decent results.
>
> **3. Contribution in the experimental results**
>
> As elaborated on in answer to your first concern, our main contributions in terms of the experimental evaluation are: (1) we provide results for the first graph-related application of a model with a symmetric cross-correlation-based cost function, (2) we remove the projector network, which both reduces the number of model parameters, as well as improves the training time, (3) we evaluate the batched scenario in self-supervised graph learning (both for our method and BGRL), (4) we show that the loss parameter can be set to λ=1/d and performs well in the graph domain (according to Tsai et al. 2021).
>
> **4. Ablation study**
>
> We would like to underline that we actually provided an ablation study. We made it while evaluating the hypothesis that using a single set of augmentation function hyperparameters is enough for our model to achieve decent results. Moreover, such a setting substantially reduces the hyperparameter search space, resulting in faster model training and selection.
>
> The choice of the actual augmentation functions and the encoder networks was motivated by our subgoal to compare against the BGRL method - to provide a fair comparison we re-used the same functions and encoders.
>
> We are willing to perform additional ablation studies that will present the influence of the augmentation functions as well as the encoder architecture. We will provide the results of this study in a few days after completing the computational part.
>
> **5. Could you please elaborate more on the remark concerning “pre-trained dataset used in the experiment”? We didn’t understand what the remark stands for.**
>
> Yours sincerely,
>
> Anonymous authors

---

> ### Author Response · Authors · 2021-11-18
> **Additional ablation study results**
>
> Dear Reviewer,
> we would like to provide an update on the ablation study experiments we mentioned in the previous answer. We decided to to extend the ablations by considering different augmentation scenarios and encoder architectures. In particular, we examined:
>
> 1) **for augmentation**:
> - using both augmentation functions (node feature masking and edge dropping) as in the original version of our method
> - using only node feature masking
> - using only edge dropping
> - without any augmentation functions
>
> 2) **for encoder architectures**:
> - MLP-based encoder (in contrast to graph neural networks)
> - one-layer GCN encoder
> - two-layer GCN encoder - as in the original version of our method
> - three-layer GCN encoder
>
> We conducted the experiments using the WikiCS dataset. We trained each model version for 1000 epochs (with 100 warmup epochs). All the remaining hyperparameters were borrowed from the best performing G-BT model on the WikiCS dataset, as reported in our paper.
>
> We visualize the influence of both the augmentation functions and encoder architectures in the attached figure. Let’s notice that using both augmentation functions provides the best results. However, using only node feature masking already leads to decent results (1pp difference; 75.9% acc). We conclude that node feature masking is more expressive than edge dropping, as using only edge dropping provides a smaller results boost (72% accuracy). Without any augmentation functions, we observe a noticeably lower quality (67% accuracy).
>
> For the encoder architectures, we notice that using the two-layer GCN model (as evaluated in our main experimental pipeline) leads to the best results. One might expect a larger model (three-layer GCN) to work better, yet we observe a performance drop when using such an encoder. This may be related to the oversmoothing issue in Graph Neural Networks. The one-layer GCN is also a reasonable choice in comparison to the two-layer model (1.1pp difference). Ignoring the graph structure by using an MLP-based encoder leads to results of about 68% accuracy.
>
> Details:
> - the MLP-based encoder is a three-layer model with batch normalization and PReLU activations after the first two layers, whereas the output of the third linear layer is unmodified (aligned with other models in our paper)
> - in the case of the one layer GCN, we used a single GCN layer without any normalization or activation functions
> - the two and three-layer GCN models are the same ones we describe in our paper
>
> We would appreciate it if the reviewer would revise the assessment of the paper given the provided responses.
>
> Yours sincerely,
>
> Anonymous authors
>
>
>
> Figure URL: https://imgur.com/a/VpGVI2a

---

### Official Review · Reviewer_H4MK · 2021-10-31

**Correctness:** 3
**Technical Novelty And Significance:** 3
**Empirical Novelty And Significance:** 3
**Recommendation:** 6
**Confidence:** 5

**Main Review:**

Disclosure: I have reviewed a previous version of this paper. The authors have included thorough full-batch experiments as well as larger scale datasets such as ogbn-products, which is much appreciated.

The paper is clear, well-written and easy to follow. The authors propose a simple but meaningful extension of the Barlow Twins idea to the graph domain, and demonstrate its effectiveness on relevant experiments. I think allowing for symmetric loss is a very important direction for graph representation learning, and the proposed solution is an elegant way of achieving that.

My main concern would be with the reported BGRL results on ogbn-products. I understand that the authors have ran BGRL under the same computational budget as G-BT, but it appears clear that BGRL needs more time to reach peak performance. Would it be possible, just to avoid muddying the waters for future work, to run BGRL for longer and report how the performance is affected? It is OK if this number is higher than G-BT's reported performance -- the authors are optimising for a different metric.

**Summary Of The Paper:**

The authors study symmetric self-supervised graph representation learning without negative samples, inspired by the Barlow Twins method previously proposed in the image domain. They illustrate that using their method, it is possible to achieve competitive performance to state-of-the-art methods such as BGRL, at a fraction of the training cost.

**Summary Of The Review:**

I think that sufficiently many of my previous concerns have been addressed, and I am now leaning on the side of acceptance. The authors have presented a useful extension of Barlow Twins into the graph domain, and now have experiments in support of the industrial relevance of their method. The novelty is somewhat limited (as is the case for most of the recent graph SSL papers that adapt image domain techniques) but it is useful in and of itself that the gains observed in images transfer well to the irregular domains.

---

> ### Author Response · Authors · 2021-11-18
> **Results of longer trained BGRL**
>
> Dear Reviewer,
>
> thank you for your review. We would like to address your concern with the experimental evaluation of the OGB Products dataset. Indeed, we limited the number of epochs to 100 for both BGRL and G-BT. However, we noticed that our model keeps improving with more epochs, which is not the case for BGRL. We additionally evaluated BGRL for 10 times more epochs (i.e., 1000 epochs) and present the results in the attached figure. As you can notice, further training of BGRL degrades the performance.
>
> We would appreciate it if following the results, the reviewer would become more convinced about the maturity of our proposed method’s performance.
>
> Yours sincerely,
>
> Anonymous authors
>
> Figure URL: https://imgur.com/a/cIf2evK

---

### Official Review · Reviewer_Bwpq · 2021-11-03

**Correctness:** 4
**Technical Novelty And Significance:** 1
**Empirical Novelty And Significance:** 1
**Recommendation:** 3
**Confidence:** 5

**Main Review:**

Strenghts:

1. Paper is clearly written and easy to follow.
2. The experimental evaluation is robust and experimental details are clearly stated.

Weakness:

The paper is the straightforward extension of previous work Barlow-Twins for graph structured data. The paper does not has any technical novelty in my opinion. I am willing to increase the score of the paper if in the rebuttal, the authors can clearly state the novelty of the paper w.r.t to the Barlow- Twin paper.

**Summary Of The Paper:**

The paper applies the recently proposed self-supervised learning method Barlow-Twins to graph structured data. For constructing the augmented version of a graph, previous methods such as edge-dropping or feature masking are used. The paper conducts experimental evaluation on datasets of various scales on both transductive and inductive setting.

**Summary Of The Review:**

Although the paper is clearly written and many experiments are presented, the paper does not meet the bar of the top conference such as ICLR because of it being a very direct application of a previous paper.


My review is rather short for this paper because based based on the lack of novelty of this paper, I do not have many questions to ask or suggestions to make.

---

> ### Author Response · Authors · 2021-11-12
> **Novelty of the paper (1/2)**
>
> Dear Reviewer,
>
> thank you for your review. We would like to address your concerns and clearly state the novelty of our paper w.r.t. to the previous Barlow Twins paper below.
>
> The only way to realistically access the paper’s novelty, in this case, is to compare it with the original Barlow Twins paper. The comparison should consider the research question, methodology, and results of both articles.  The research questions are different – we wanted to answer whether one could utilize a symmetric negative-sample free loss function to obtain expressive embeddings at a fraction of the training time of state-of-the-art graph SSL methods. The methodology seems similar, as it appears in all graph SSL papers nowadays. The obtained results are aligned to the nature of different problems that are solved.
>
> Due to our best knowledge, our paper is a first attempt to utilize a symmetric cross-correlation-based cost function (currently known under the name “Barlow Twins” loss) in the graph representation learning domain.
>
> Let us discuss in detail all new significant proposals that differentiate it from the previous work:
> “Barlow Twins: Self-Supervised Learning via Redundancy Reduction.” In general, the similarity to that work is limited to the utilized cost function. All other computational steps in our proposal are different due to another class of objects we aim at obtaining representation. We introduced a new appropriate encoder, data augmentation functions, simplified part of neural network structure that is not necessary to apply in graph case, and provided experimental results for the batched scenario.
>
> **1. Network encoder structure and its relevancy to the graph data**
>
> The original paper was dealing with vision data. In this domain, any proposed learning model should be invariant with respect to the underlying translational symmetry group (e.g., in image classification, the class should not depend on the object location on that image). In the case of the original Barlow Twins method, this issue was solved by employing the ResNet architecture (in fact, using convolutional layers with pooling operations). In the case of graphs, we care about invariance to an entirely different symmetry group, namely the permutation group (of the node ordering). We implement it using the Graph Convolution and Graph Attention encoder models. Yet it is not automatically known whether such encoder swap would work and provide meaningful representations in the first place. Moreover, in terms of images, we generally assume that each data point (image) is independent (known as the IID property). In the case of graphs, nodes (which we learn representations for), are dependent on other nodes (via edges). Using the batched setting from the original Barlow Twins method might be straightforwardly applicable in our case, as we are passing the whole graph (or a subsampled batch). Nevertheless, our experiments provided more insight to that matter and turned out that it is possible.
>
> **2. Augmentation functions**
>
> The original Barlow Twins paper proposed using a set of well-known augmentation functions for the image domain, i.e. “random cropping, resizing to 224 × 224, horizontal flipping, color jittering, converting to grayscale, Gaussian blurring, and solarization”. None of these is applicable to the graph domain. We had to find suitable augmentation functions and evaluate them. The space of such functions is vast, but as stated in our paper, we wanted to compare our proposed method to BGRL. Hence we used the same augmentation functions as in their paper, namely, edge dropping and node feature masking.
>
> Another modification in our paper is related to the probabilities of the augmentation functions. Both the original Barlow Twins and BGRL papers propose to utilize different probabilities to generate both augmented data views. We proposed to simplify the hyperparameter space and verified experimentally that it is enough to use the same probabilities to generate both views. Such a setting significantly reduces the hyperparameter space search and provides the same results.
>
> **3. Simplification of the model structure**
>
> Original Barlow Twins method for images required a projector network to deal with the dimensionality of the ResNet encoder output (2048). Surprisingly, the method kept improving with an increase of projector network dimensionality. In our proposed architecture, we proposed to completely omit the projector network and work directly with the embedding vectors computed by the encoders. In preliminary experiments, we verified that the use of a projector network does not improve the downstream task results. Moreover, removing the projector network noticeably reduces the number of parameters and learning time.

---

> ### Author Response · Authors · 2021-11-12
> **Novelty of the paper (2/2)**
>
> **4. Full-batch vs Batched processing**
>
> A popular strategy in graph representation learning is processing the whole graph at once (full-batch). This limits the applications of GNNs to small and medium-sized graphs. Nevertheless, our main competitor BGRL utilized only this scenario in their experiments. In our case, we wanted to show the expressiveness of our proposed model for large-scale graphs (ogb-products dataset). We proposed to adopt a graph batching strategy to our model (through neighbor sampling strategy). Moreover, we reproduced BGRL for the batched scenario.
>
> **5. Novelty in experimental results**
>
> Our experiments provide results for the first graph-related application of a model with a symmetric cross-correlation-based cost function. Moreover, it empirically confirms that results for node classification tasks with λ=1/d perform well (according to Tsai et al. 2021).
>
> We strongly believe that all the provided arguments will convince the reviewer to change the reception of the paper and will consider the change of the paper’s assessment. We stand for the fact that it is hardly possible to obtain better tradeoff results (time-accuracy), as in our paper, without introducing any novelty to the ones previously published.
>
> **Summary**
>
> In summary, our contribution compared to Barlow Twins original paper includes: a new appropriate encoder, data augmentation functions, and a simplified part of neural network structure that is not necessary in the graph case and provide experimental results for the batched scenario.  All of the mentioned differences will be highlighted in the revised version of the manuscript.
>
> Yours sincerely,
>
> Anonymous authors

---

### Decision · Program_Chairs · 2022-01-20

**Decision:**

Reject

**Comment:**

The paper proposes to use the recently introduced "Barlow-twins" contrastive learning objective, to the case of graph networks. The main concern raised by reviewers was the limited novelty of this work, which they argued mostly combines existing lines of work, and does not introduce sufficiently new concepts. This was also discussed between the authors and the reviewers.
Having read the paper and the reviews, I tend to agree with the reviewers that this paper is more of a combination of existing works, and their relatively straightforward application to the graph network domain. Thus, although the empirical results are encouraging, I agree the paper has limited novelty, and falls below the ICLR acceptance bar.